# Suppressing phase disproportionation in quasi-2D perovskite light-emitting diodes

Kang Wang[1], Zih-Yu Lin [1], Zihan Zhang[2], Linrui Jin[3], Ke Ma [1], Aidan H. Coffey[1], Harindi R. Atapattu[4], Yao Gao [1], Jee Yung Park [1], Zitang Wei[1], Blake P. Finkenauer [1], Chenhui Zhu[5], Xiangeng Meng[6], Sarah N. Chowdhury[7], Zhaoyang Chen[8], Tanguy Terlier [9], Thi-Hoai Do[10], Yan Yao [8], Kenneth R. Graham [4], Alexandra Boltasseva [7], Tzung-Fang Guo [10], Libai Huang [3], Hanwei Gao[2], Brett M. Savoie [1] & Letian Dou [1,7] ✉

Electroluminescence efficiencies and stabilities of quasi-two-dimensional halide perovskites are restricted by the formation of multiple-quantum-well structures with broad and uncontrollable phase distributions. Here, we report a ligand design strategy to substantially suppress diffusion-limited phase disproportionation, thereby enabling better phase control. We demonstrate that extending the π-conjugation length and increasing the cross-sectional area of the ligand enables perovskite thin films with dramatically suppressed ion transport, narrowed phase distributions, reduced defect densities, and enhanced radiative recombination efficiencies. Consequently, we achieved efficient and stable deep-red light-emitting diodes with a peak external quantum efficiency of 26.3% (average 22.9% among 70 devices and cross-checked) and a half-life of ~220 and 2.8 h under a constant current density of 0.1 and 12 mA/cm², respectively. Our devices also exhibit wide wavelength tunability and improved spectral and phase stability compared with existing perovskite light-emitting diodes. These discoveries provide critical insights into the molecular design and crystallization kinetics of low-dimensional perovskite semiconductors for light-emitting devices.

Two-dimensional (2D) perovskites adopt the general chemical formula of $L_2A_{n-1}B_nX_{3n+1}$, where A is a small cation $(CH(NH_2)_2^+, FA^+)$, B is a divalent metal cation $(Pb^{2+})$, X is a halide anion $(Br^-$ or $I^-)$, L is a large organic ligand such as widely used butylammonium (BA), and $n$ is the number of perovskite layers in the 2D structure (Fig. 1a). They are emerging candidates for next-generation light-emitting diodes (LEDs) because of their outstanding optoelectronic properties with remarkable tunability[1–9]. Impressive progresses have been made and the external quantum efficiencies (EQE) of quasi-2D perovskite-based LEDs $(n > 2)$ have recently exceeded 20% in both green and near-infrared regions[10–17]. The efficiency and stability are likely limited by the low phase purity and high defect densities of typical 2D perovskite films. Although an average $n$ number is controlled by the stoichiometry of the precursor solution, a broad phase distribution ranging

[1]Davidson School of Chemical Engineering, Purdue University, West Lafayette, IN, USA. [2]Department of Physics, Florida State University, Tallahassee, FL, USA. [3]Department of Chemistry, Purdue University, West Lafayette, IN, USA. [4]Department of Chemistry, University of Kentucky, Lexington, KY, USA. [5]Advanced Light Source, Lawrence Berkeley National Laboratory, Berkeley, CA, USA. [6]School of Materials Science and Engineering, Qilu University of Technology (Shandong Academy of Sciences), Jinan, China. [7]Birck Nanotechnology Center, Purdue University, West Lafayette, IN, USA. [8]Department of Electrical and Computer Engineering and Texas Center for Superconductivity at the University of Houston (TcSUH), University of Houston, Houston, TX, USA. [9]SIMS laboratory, Shared Equipment Authority, Rice University, Houston, TX, USA. [10]Department of Photonics, National Cheng Kung University, Tainan, Taiwan. ✉e-mail: dou10@purdue.edu

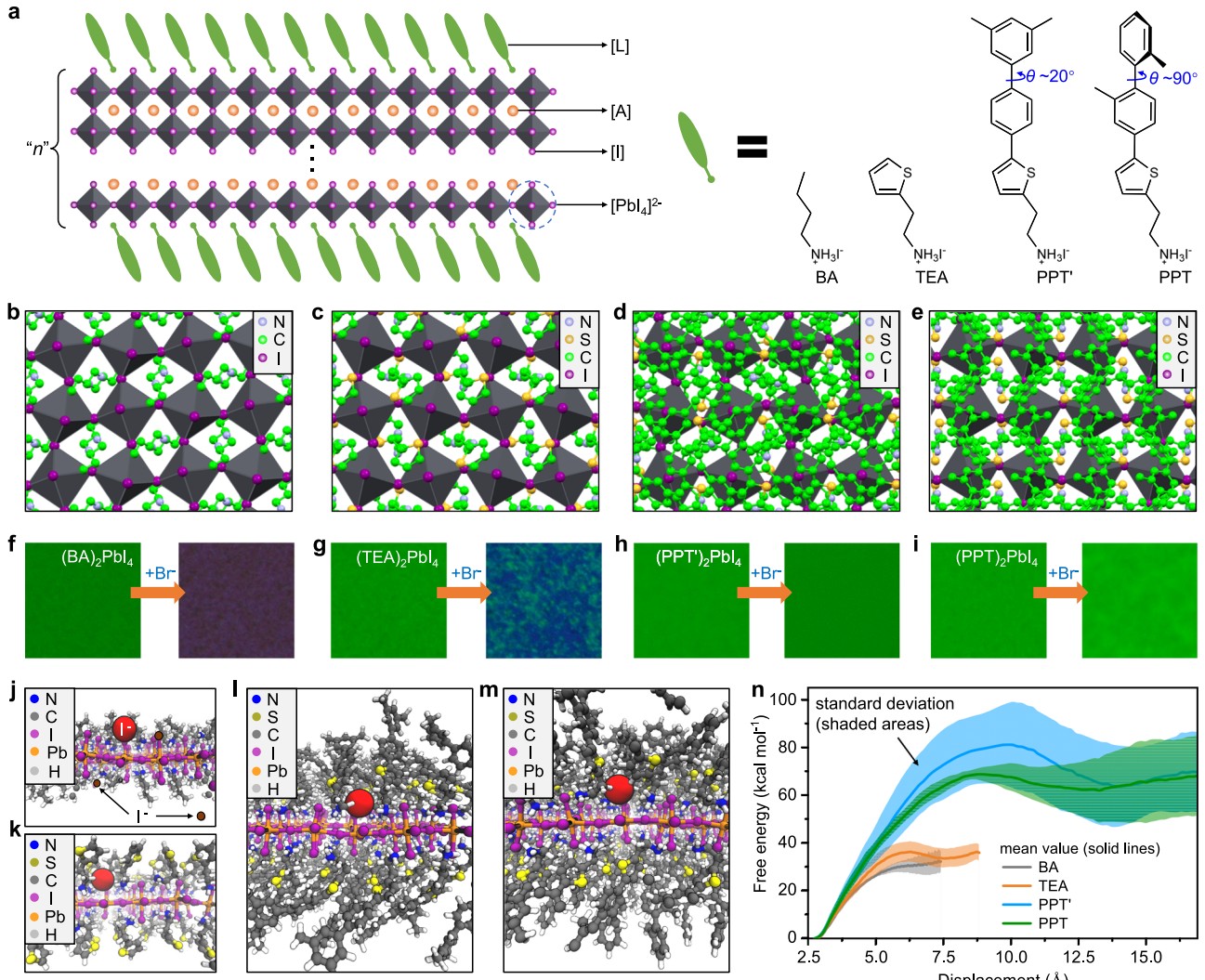

**Fig. 1 | Blocking ionic transport in 2D perovskites by bulky organic ligands.**
**a** The schematic illustration for the general structure of quasi-2D perovskites and the chemical structures of large organic ligands, BA, TEA, PPT′, and PPT, studied in this work. The two methyl groups at PPT ligand were designed to induce steric hindrance and break the planarity of the ligand, thus leading to an increased dihedral angle $\theta$ of ~90°. **b**–**e** The top view of the single crystal structure of $(BA)_2PbI_4$ (**b**), $(TEA)_2PbI_4$ (**c**), $(PPT')_2PbI_4$ (**d**), and $(PPT)_2PbI_4$ (**e**). Black diamonds represent inorganic octahedral $[PbI_4]^{2-}$ layers; the disorder within $(PPT)_2PbI_4$ structure and all hydrogen atoms are omitted for clarity. **f**–**i** PL images of $n = 1$ 2D perovskite thin films before (left) and after (right) HBr vapor treatment. **j**–**m**, Snapshots from the MD simulations for $(BA)_2PbI_4$ (**j**), $(TEA)_2PbI_4$ (**k**), $(PPT')_2PbI_4$ (**l**), and $(PPT)_2PbI_4$ (**m**), showing a side view of the crystal structure after steering out $I^-$ from the lattice. The red spheres denote the magnified $I^-$ being pulled while the brown spheres in (**j**) denote $I^-$ that breaks free. The perspective is chosen to clearly uncover the ion diffusion pathways. **n** Free energy for halide diffusion from an apical position through the organic ligand layer to a neighboring perovskite layer. The solid lines and shaded areas indicate the mean and standard deviations calculated from five trajectories, respectively.

from $n = 1$ to $n = \infty$ is usually formed after thin film processing due to kinetic and thermodynamic factors. Such a broad $n$-distribution in quasi-2D perovskites could be understood as a result from phase "disproportionation" of an initial $n$-value material to yield lower and higher $n$-phases, which leads to phase broadening and instability over time. In addition, films with mixed phases exhibit cascade energy transfer from low-$n$ to high-$n$ phases[2,3], resulting in emission that is dominated by the composition of higher-$n$ phase and ultimately difficult to control. A few recent efforts have been devoted to regulating phase distribution and smoothing the energy landscape in quasi-2D perovskite thin films[18–21], but has exhibited limited success in wavelength-tunable LEDs due to ineffective phase control.

## Results

### Blocking ionic transport by ligand design

Phase disproportionation of intermediate $n$-values is spontaneous for typical 2D perovskites because the $n = 1$ phase is enthalpically favored

and a mixture of $n$-phases is entropically favored[18]. Since mass transport (ion diffusion) must happen between layers during phase disproportionation, we hypothesized that the ligands might control the kinetics by modulating either the interlayer diffusion barrier or the number of interfacial defects through which ion diffusion can occur[17]. Recent works have demonstrated that conjugated ligands (e.g., thiophenylethylammonium, TEA, Fig. 1a) can inhibit ion diffusion within and between 2D perovskite heterostructures[22–24], which provides the means of testing the above hypothesis. Here, we further designed and synthesized two novel organic conjugated ligands, PPT′ (2-(5-(3′,5′-dimethyl-[1,1′-biphenyl]−4-yl)thiophen-2-yl)ethyl-1-ammonium iodide) and PPT (2-(5-(2,2′-dimethyl-[1,1′-biphenyl]−4-yl)thiophen-2-yl)ethyl-1-ammonium iodide) (Fig. 1a). Their synthesis procedures are detailed in Supplementary Information[25]. The design of these ligands was motivated by their relatively large π-systems, which is already an established factor for suppressing ion-diffusion, and their distinct cross-sections due to the increased steric barrier associated with the

phenyl-phenyl dihedral of PPT. This latter factor may lead to better solution processability and unusual phase control behaviors, which is yet to be explored.

To examine these hypotheses, we synthesized $n = 1$ lead iodide-based 2D perovskite bulk crystals for PPT' and PPT, namely (PPT')$_2$PbI$_4$ and (PPT)$_2$PbI$_4$. The top views of their single crystal structures are shown in Fig. 1b–e with previously reported (BA)$_2$PbI$_4$[26] and (TEA)$_2$PbI$_4$[27] as references (see Supplementary Figs. 1, 2 and Supplementary Table 1 for more details). The TEA organic layer exhibits slightly increased surface coverage on the PbI$_4^{2-}$ inorganic octahedron layers relative to BA, while the PPT' and PPT layers provide significantly better coverage. To test their stability against ion diffusion, $n = 1$ 2D perovskite thin films were treated with hydrobromic acid vapor. The PL image showed that the emission of (BA)$_2$PbI$_4$ completely converted from green to purple (Fig. 1f and Supplementary Fig. 3a), implying that Br$^-$ easily penetrates the BA ligand layer and substitutes the I$^-$ in the inorganic layer. In the case of (TEA)$_2$PbI$_4$, the film exhibited mixed green and blue emissions due to partial substitution of I$^-$ by Br$^-$ (Fig. 1g and Supplementary Fig. 3b). Surprisingly, neither the (PPT')$_2$PbI$_4$ or (PPT)$_2$PbI$_4$ film exhibited a PL color change (Fig. 1h, i and Supplementary Fig. 3c, d), suggesting minimal Br$^-$ penetration. Note, there is less blue shift in the PL spectra of (PPT)$_2$PbI$_4$ compared with (PPT')$_2$PbI$_4$ after treatment (Supplementary Fig. 3c, d), indicating slightly better protection by PPT ligand. The overall ligand-dependence of ion penetration agrees with the crystal-based surface coverage analyses.

To provide molecular insight into these distinct behaviors, molecular dynamics (MD) simulations were used to compare the free energy profiles of interlayer I$^-$ diffusion in perovskites substituted with different ligands. Phase disproportionation necessarily involves the exchange of small A-cations, halide anions, and lead cations between layers, either through defects, layer edges, or through direct diffusion across the organic ligand layer. The activation energies of the latter mechanism are compared as a starting point for interpreting the distinct behaviors of these ligands. Umbrella sampling was used to pull an I$^-$ out of its crystal lattice to diffuse through the organic ligand layer of model $n = 1$ (BA)$_2$PbI$_4$, (TEA)$_2$PbI$_4$, (PPT')$_2$PbI$_4$, and (PPT)$_2$PbI$_4$ systems. In each simulation, periodic boundary conditions were used to effectively simulate stacked 2D perovskites with an example shown in Supplementary Fig. 4. The crystal structures of (BA)$_2$PbI$_4$ and (TEA)$_2$PbI$_4$ became distorted during the pulling process (Fig. 1j, k, also see top view in Supplementary Fig. 5), resulting in several neighboring ions being liberated to diffuse between layers (brown spheres in Fig. 1j). In contrast, (PPT')$_2$PbI$_4$ and (PPT)$_2$PbI$_4$ retained their crystal structures during ion diffusion (Fig. 1l, m, also see top view in Supplementary Fig. 5). This comparison provides qualitative evidence that the bulkier ligands stabilize the perovskite lattice even in the presence of ion defects. The free energy barrier for diffusion of I$^-$ through the ligand increases with length and π-conjugation such that BA < TEA < PPT' ~ PPT (Fig. 1n). The free energies required for PPT' and PPT are twice those of BA and TEA, indicating that these bulky organic conjugated ligands can effectively inhibit direct interlayer ion diffusion and potentially also hinder diffusion limited phase disproportionation. Notably, the smaller interstitial cations and Pb$^{2+}$ also need to undergo interlayer diffusion during the phase disproportionation of high-$n$ phases. Although we expect similar trends for these ions, the simulations of $n > 1$ systems are currently beyond the scope of this work. Regardless, the $n = 1$ simulations could provide a rationale for kinetically limited disproportionation via ligand design.

## Phase distribution control and mechanistic understanding

To assess the impact of these ligands on phase distribution, they were used to fabricate lead iodide-based quasi-2D perovskite ($n > 1$) thin films that were subjected to multi-mode characterizations. As shown in Fig. 2a, the thin films produced from BA and TEA ligands using a stoichiometric ratio of nominal $\langle n \rangle = 3$ (i.e., L$_2$FA$_2$Pb$_3$I$_{10}$ in precursor

solution, L = BA or TEA here) show absorption peaks at 572 and 776 nm, which correspond to $n = 2$ and $n \sim \infty$ (3D) phases, respectively. The PPT'-based film exhibits distinct absorption peaks at 572, 633, and 682 nm corresponding to $n = 2$, 4, and 5 phases, respectively. The PPT-based film exhibits absorption peaks only at 633 and 682 nm, suggesting predominant $n = 4$ and 5 phases with a narrower phase distribution. The photoluminescence (PL) of each film was also characterized (Fig. 2b), where multiple broad emission peaks from $n = 2$ and other high-$n$ phases are evident in the BA, TEA, and even PPT' films. Interestingly, the PL emission profile of the PPT film is well-defined and dominated by $n \approx 4$–6 phases. The phase distributions of each film were estimated by linear superposition (Fig. 2c) using the absorption coefficient of each $n$-phase (Supplementary Fig. 6). The narrow phase distribution in the PPT film leads to blue shift of the main emission peak from near infrared to red (Fig. 2d) and improved PL quantum yield (PLQY) (Supplementary Fig. 7).

Grazing incidence small-angle X-ray scattering (GISAXS) was employed as a structural probe to quantify the different $n$ phases in the films. Based on the out-of-plane scattering patterns, the BA and TEA films are primarily composed of a high-$n$ (near 3D) phase accompanied by a smaller fraction of low-$n$ phases (Fig. 2e, f). The PPT' film mainly consists of median-$n$ phases ranging from $n = 2$ to 8 (Fig. 2g), whereas the PPT film is dominated by a narrower phase distribution ranging from $n = 2$ to 6 (Fig. 2h). These data comport with the optical measurements and together provide strong evidence of effective phase distribution control using the PPT ligand, and to a lesser degree the PPT' ligand. This interpretation that suppressing off-target disproportionation by PPT' and PPT ligands is also corroborated by other scattering and microscopy characterizations such as powder X-ray diffraction in Supplementary Fig. 8 and Kelvin probe force microscopy (KPFM) contact potential difference (CPD) images[28] in Supplementary Fig. 9. As an additional demonstration, perovskite thin films were fabricated with a series of other organic conjugated ligands (such as PEA, 2 P, 2 T, and 3 T, see Supplementary Fig. 10), all of which were confirmed to exhibit broad $n$ distributions that are unsuitable for efficient and color tunable LEDs. These observations highlight the unique role of the newly developed ligands in controlling thin film growth and phase formation.

To visualize the phase disproportionation process and investigate the underlying mechanism, ex-situ PL during spin-coating, in-situ PL during thermal annealing and the corresponding GISAXS measurements were conducted. It is revealed that $n \sim 3$ phases form at the initial spinning stage for all cases (Supplementary Figs. 11, 12), but BA and TEA cations lead to phase disproportionation to low- and high-$n$ phases during late-stage of spinning, whereas PPT' and PPT films retain $n \sim 3$ phase throughout the spinning process (Supplementary Fig. 11). Then, a short-term thermal annealing (100 °C for 10 min) was applied and more thorough phase disproportionation occurs in BA and TEA based films, while PPT' and PPT based films are relatively stable (see Supplementary Figs. 13–17 for more detailed discussions). However, upon heating the PPT film at 100 °C for a long period (> 5 h), small amounts of $n = 2$ and $n \sim \infty$ phases appeared (Supplementary Fig. 18), validating that the median-$n$ phases are kinetic products and that they can eventually disproportionate, albeit with a higher activation energy than the other films. These results are also in good agreement with that of other characterizations, such as ex-situ UV-vis (Supplementary Fig. 19) and in-situ grazing incidence wide-angle X-ray scattering (GIWAXS) (Supplementary Fig. 20). We further carried out disproportionation reaction kinetics study on the spin-coated wet films by following a reported method[29]. The activation energy retrieved from this study (Supplementary Figs. 21–25) is consistent with the trend of simulated free energy for ion diffusion (Fig. 1n). These observations can be summarized as a crystal formation model for quasi-2D perovskites, where ligands play an important role in regulating interlayer ion-transport and ultimately the kinetics of phase

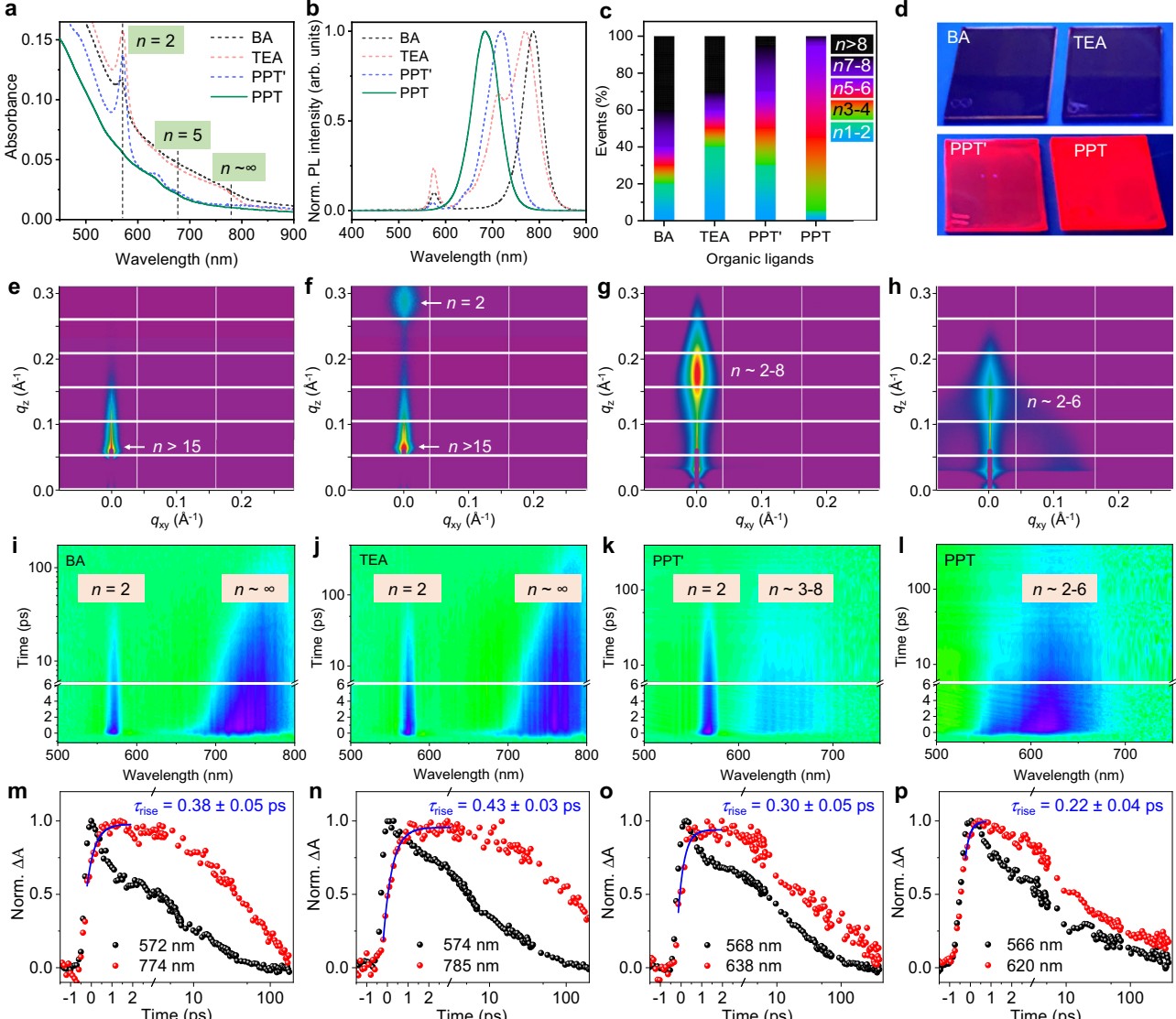

**Fig. 2 | Phase distribution control and energy transfer dynamics of quasi-2D perovskite thin films. a, b** UV-vis (**a**) and PL (**b**) spectra of the quasi-2D perovskite films. From left to right: the vertical dash lines in (**a**) indicate the absorption peaks of the $n = 2$, $n = 5$, and $n \sim \infty$ (3D) phases. All the films were fabricated from the precursor solutions with a stoichiometric ratio of nominal $<n> = 3$. **c** The relative contents of different $n$ phases for perovskite films. The relative contents were estimated from the absorption spectra based on the linear superposition of the corresponding absorption coefficient of each $n$-phase. This can be further verified by the GISAXS and TA data shown below. **d** Photographs of quasi-2D perovskite films with different organic ligands under ultraviolet lamp irradiation. **e−h** GISAXS pattern of (**e**) BA, (**f**) TEA, (**g**) PPT', and (**h**) PPT based films. $q_{xy}$ and $q_z$ represent the in-plane and out-of-plane scattering vectors, respectively. The dominated $n$ phases in the films were determined by referring to their corresponding single-crystal structures. **i−l** Pseudo color maps of TA spectra and **m−p**, corresponding kinetics curves at two different wavelengths of (**i, m**) BA, (**j, n**) TEA, (**k, o**) PPT', (**l, p**) PPT based films. The energy transfer time constant is fitted to be $0.38 \pm 0.05$, $0.43 \pm 0.03$, $0.30 \pm 0.05$, and $0.22 \pm 0.04$ ps for BA, TEA, PPT', and PPT film, respectively.

disproportionation (Supplementary Fig. 26). Interestingly, the control over the $n$-phase provided by the PPT ligand enables us to tune the PL emission wavelength from 610 to 720 nm by adjusting the stoichiometric ratio of precursor solutions (Supplementary Figs. 27, 28) or annealing temperature (Supplementary Fig. 29), making them useful in stable and color-tunable LEDs.

Transient absorption (TA) spectroscopy measurements were then performed to investigate the energy transfer dynamics between different $n$ phases. The distinctive photo bleach peaks were assigned to different $n$-phases in the quasi-2D perovskite films (Fig. 2i–l), which again aligns with our analysis on phase distribution. The corresponding TA dynamics demonstrate the energy transfer from low-$n$ to high-$n$ phases among all the films, by comparing the rising time of extracted kinetic curves at different wavelengths (Fig. 2m–p). Accordingly, the energy transfer time constant in the PPT film is measured to be ~0.2 ps,

which is faster than that in other films (~0.3 to 0.4 ps)[30]. The faster energy transfer processes may be associated with a combination of fewer transfer steps (better phase purity) and reduced carrier-trapping (lower defect density) in the PPT film[10].

## Improved optoelectronic properties by phase control
Decreasing defect density and suppressing non-radiative recombination pathways are critically important for LEDs. Figure 3a illustrates cascade energy transfer between different $n$-phase domains in perovskite films. The energy transfer is usually accompanied with trap-related nonradiative losses and undesired radiative loss if inter-phase energy transfer is less efficient. This suggests that cutting down transfer steps will be beneficial to reduce those losses, thus boosting the PLQY. By gradually changing the excitation wavelength to lower energy (longer wavelength) to decrease the absorption of the low $n$

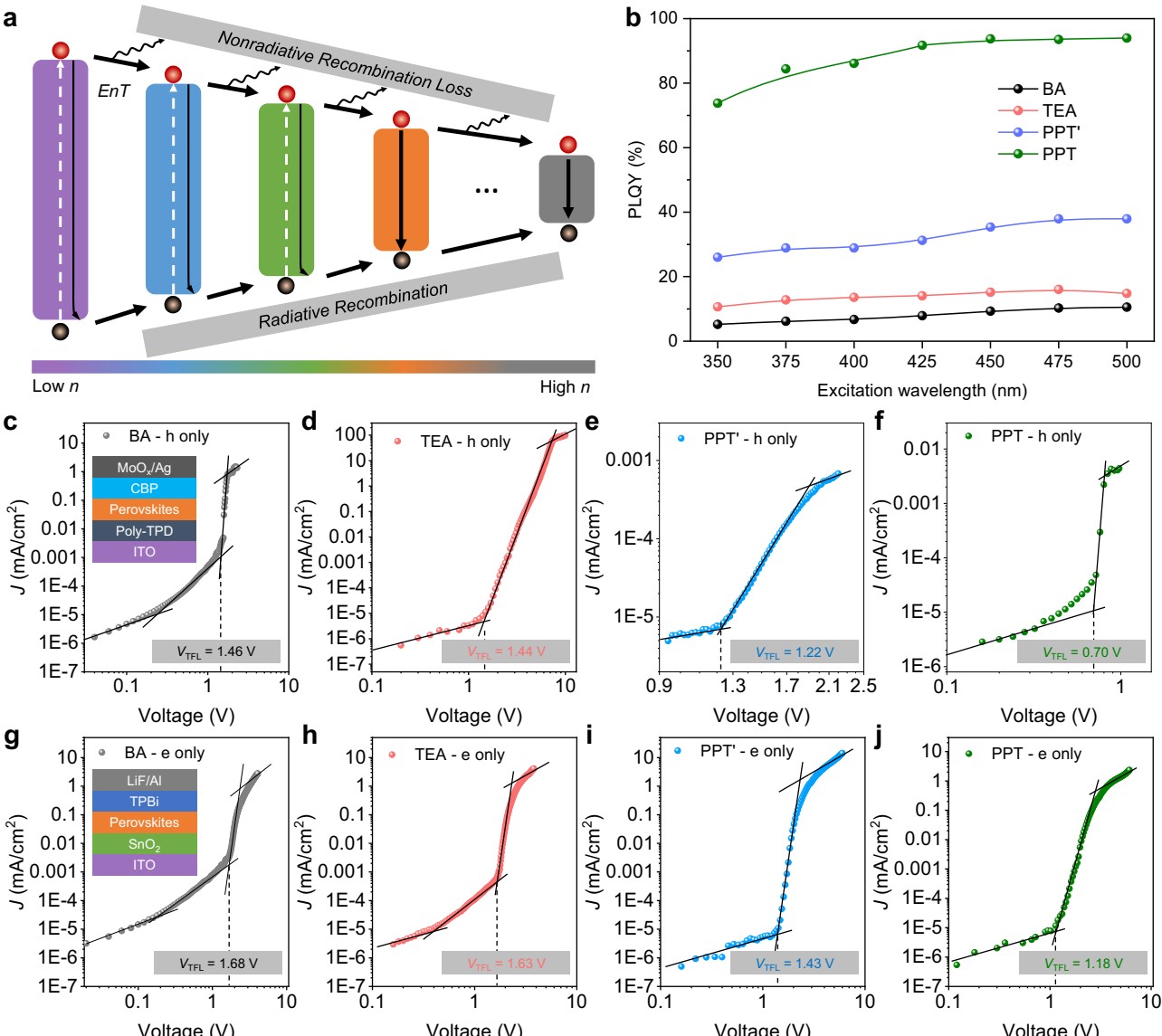

**Fig. 3 | Improved optoelectronic properties with better phase control.**
**a** Schematic illustration of cascaded energy transfer between different *n*-phases. The energy transfer (*EnT*) in quasi-2D perovskites is usually accompanied with nonradiative losses and undesired radiative recombination losses if without effective phase distribution control. **b** Excitation wavelength dependent PLQY of BA (black), TEA (red), PPT' (blue), and PPT (green) based quasi-2D perovskite films. **c**–**j**, Current density versus voltage curves for hole-only (**c**–**f**) and electron-only

(**g**–**j**) devices under dark conditions. Inset shows the device structures. ITO, indium tin oxide; Poly-TPD, Poly(N,N'-bis-4-butylphenyl-N,N'-bisphenyl)benzidine); CBP, 4,4'-Bis(N-carbazolyl)−1,1'-biphenyl; TPBi, 1,3,5-tris(1-phenyl-1H-benzimidazol-2-yl) benzene. SCLC fittings (solid lines) reveal the trap-filling limit voltage (*V*~TFL~) for hole and electron, respectively, showing reduced trap densities of quasi-2D perovskites made from PPT ligand.

phases, we clearly observed enhanced PLQY for all films (Fig. 3b). We then carried out ultraviolet photoemission spectroscopy (UPS) measurement (Supplementary Figs. 30, 31). The results indicate that BA and TEA are more n-type, i.e., the Fermi energy is closer to the conduction band, which may be attributed to the presence of more defect states. Time-resolved PL (TRPL) (Supplementary Fig. 32) qualitatively revealed remarkably reduced trap states with PPT. Such a lower trap density substantially leads to a much higher PLQY (up to 94%) of PPT film compared to that of BA, TEA, and PPT' cases under all excitation conditions (Fig. 3b, Supplementary Fig. 7).

We further built hole (h)-only (inset in Fig. 3c) and electron (e)-only (inset in Fig. 3g) devices for electrical transport measurements to quantify the trap densities. Using a space-charge-limited current (SCLC) model, we estimated the trap state densities from: $N_{t(e/h)} = 2\varepsilon_r\varepsilon_0 V_{TFL(e/h)} / (qd^2)$, where $N_t$ is the trap state density, $V_{TFL}$ is the trap-filled limit voltage, $d$ is the distance between the electrodes, $q$ is the

elementary charge, and $\varepsilon_0$ and $\varepsilon_r$ are the vacuum permittivity and relative permittivity, respectively. It was found that the $V_{TFL}$ for both electron and hole are in the order of BA > TEA > PPT' > PPT (Fig. 3c–j). The hole trap densities for BA, TEA, PPT', and PPT-based devices were determined to be $4.4 \times 10^{16}$, $4.3 \times 10^{16}$, $3.7 \times 10^{16}$, and $2.1 \times 10^{16}$ cm$^{-3}$, respectively. The electron trap densities were determined to be $5.0 \times 10^{16}$, $4.8 \times 10^{16}$, $4.2 \times 10^{16}$, and $3.5 \times 10^{16}$ cm$^{-3}$, respectively. The reduced defect density along with improved PLQY make PPT-based quasi-2D perovskite thin films promising for efficient LEDs.

## LED device characteristics
Device architecture is shown in Fig. 4a[31,32]. Poly-TPD was used as hole-transporting layer and TPBi was used as electron-transporting layer. An ultrathin polyvinylpyrrolidone (PVP) interlayer was inserted in between Poly-TPD and perovskite layer to improve the wettability and mitigate the interfacial loss[12,33,34]. The thicknesses of Poly-TPD,

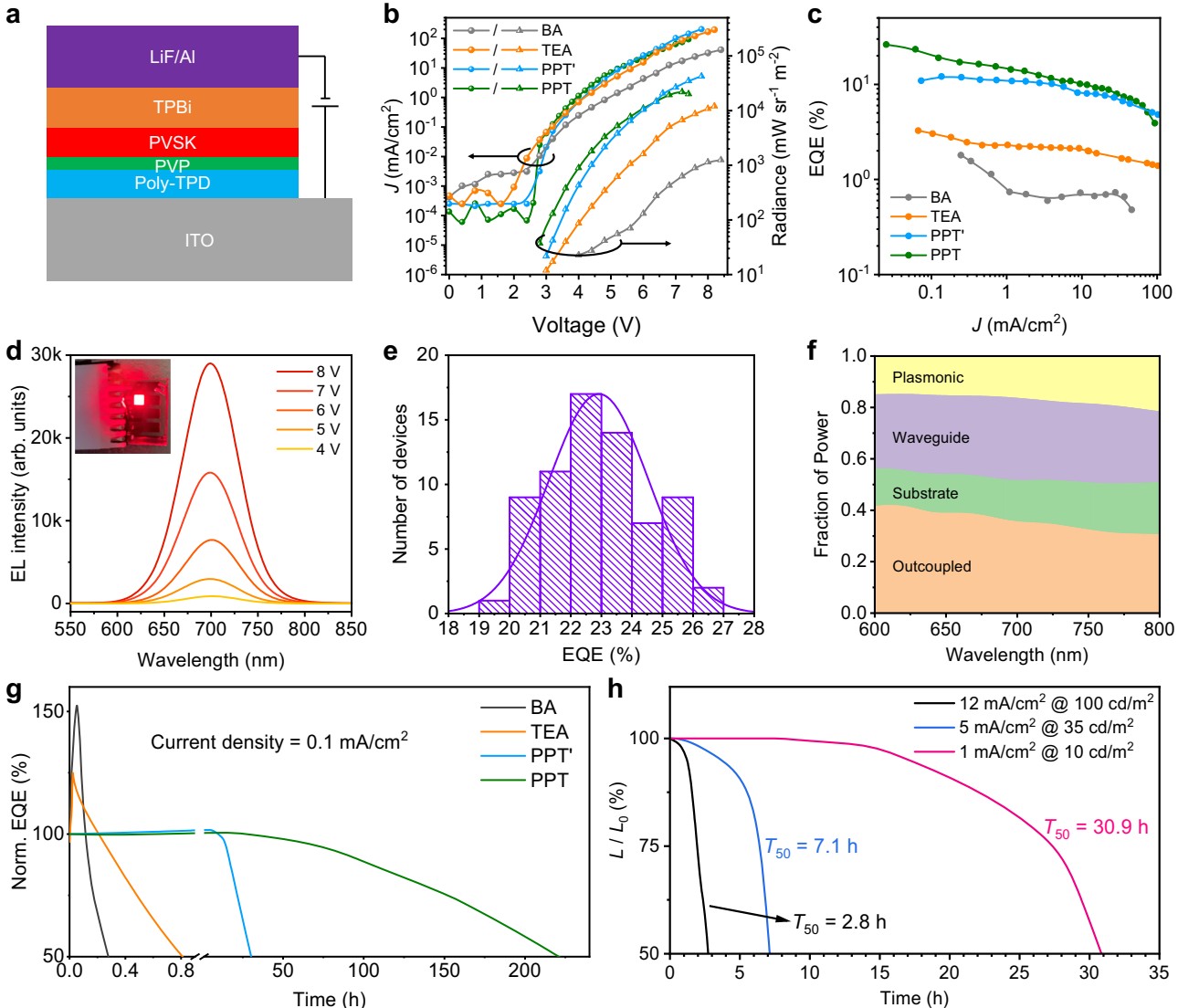

**Fig. 4 | Device performance of distribution controlled quasi-2D perovskite LEDs. a** Device architecture. PVSK indicates the quasi-2D perovskites investigated in this work. **b**, **c** Current density-voltage-radiance (*J*-*V*-*R*) curves (**b**) and EQE characteristic (**c**) of quasi-2D perovskite LEDs. **d** EL spectra under forward biases of 4, 5, 6, 7 and 8 V. Inset is a photograph of a working device. **e** Histogram of EQEs measured from 70 devices, which gives an average value of 22.9% and a relative standard deviation of 6.9%. **f** Simulated fractional power distribution in the LED structure as a function of emission wavelength. **g** Operational stability measurement of quasi-2D perovskite LEDs based on different ligands without encapsulation, performed in nitrogen filled glove box at a constant current density of 0.1 mA/cm². **h** Operational stability of PPT device under various current densities.

perovskite and TPBi layers are around 30, 20, 60 nm, respectively, as determined by optical profilometer. The scanning electron microscope (SEM) images (Supplementary Fig. 33) of quasi-2D perovskites show uniform films with complete surface coverage free of pinholes. Atomic force microscope (AFM) measurements show that the as-prepared PPT-based thin film has a root-mean-square (r.m.s.) roughness of ~0.67 nm, which is lower than thin films prepared with other organic ligands (Supplementary Fig. 9a–d) and favorable for reducing current leakage in LEDs. The current density-voltage-radiance (*J*-*V*-*R*) and EQE-current density (EQE-*J*) curves (Fig. 4b, c) of LEDs exhibit different EL behaviors. The turn-on voltage for radiance substantially decreases from 4.0 V (BA) to 2.8 V (PPT), which confirms the PPT film has reduced defect densities. To this end, the resulting EQEs for different ligands are in good agreement with their phase distribution analysis and PLQY data. Excitingly, the PPT-based device delivers a peak EQE of 26.3%, representing the most efficient red LEDs based on quasi-2D perovskite reported so far (Supplementary Table 2). The luminance curves and current efficiency for PPT' and PPT devices are

plotted in Supplementary Fig. 34. More details on device optimizations and corresponding device performances can be found in Supplementary Figs. 35–39.

Figure 4d shows the EL spectra of our champion device under different driving voltages and an EL image of an operating device (Fig. 4d, inset). The EL spectra centered at 700 nm corresponds to a deep-red emission, which is distinct from the near-infrared emission of the BA, TEA, and PPT' devices (Supplementary Fig. 40). In addition, the EL spectra do not shift with increasing the voltage, indicating improved color stability. An EQE histogram for 70 devices shows an average EQE of 22.9% with a low relative standard deviation of 6.9% (Fig. 4e), demonstrating good reproducibility. The device characteristics regarding *J*-*V*-*R* and EQE-*J* curves of all 70 devices are included in Supplementary Fig. 41. The champion device was cross-checked at National Cheng Kung University (NCKU, Taiwan), which shows a peak EQE of 22.6% (Supplementary Fig. 42). We also performed an optical simulation on our LEDs to clarify the outcoupling efficiencies by using a classical dipole model with self-retrieved permittivity for each layer

(Supplementary Fig. 43). The result suggests a light outcoupling efficiency of 35.3% for the champion device (Fig. 4f), which may lead to a theoretically predicted maximum EQE of 33% when considering a PLQY of 94%. Further optimizations can be performed including passivating defects and controlling the orientation of quasi-2D perovskites[35], which shall push the EQE to an even higher level.

Ion transport not only affects the phase distribution in quasi-2D perovskite, but also limits the device stability under electrical bias. PPT' and PPT-based devices exhibit negligible hysteresis in the forward and reverse $J$-$V$ scans compared with BA, and TEA-based devices (Supplementary Fig. 44a–d). This suggests effective immobilization of ions under electrical operation by our designed ligands. We further employed a capacitance spectroscopy technique to investigate the ion migration in the working devices (Supplementary Fig. 44e). The relative magnitudes of capacitance at low-frequency region ($<10^3$ Hz) show remarkable increase, which is attributed to the accumulation of electronic and ionic charges at the device interfaces and perovskite grain boundaries[36]. Specifically, the increase of relative capacitance magnitude at the low-frequency region is reduced following the order of BA > TEA > PPT' > PPT. The correlation between structure of the organic ligands and ionic transport in the perovskites was further verified using galvanostatic tests (Supplementary Figs. 44f–h, 45)[37]. Consistent with other theory and relevant measurements, the perovskites with BA and TEA indeed show a relatively higher ionic conductance, while the PPT' and PPT incorporated perovskites show a much lower value.

We tested operational stability for all four types of devices initially at a relatively low constant current density of 0.1 mA cm$^{-2}$ to better analyze the degradation processes (Fig. 4g). The EQE of BA based device overshot to 162% of the initial value, then dropped to 50% after a time of $T_{50}$ = 16.7 min. The main reasons behind this overshoot are still

unclear in the field but are usually attributed to the ion migration and accumulation at two interfaces of the perovskite layer. Such ion accumulation could benefit the charge injection by a strong local electric field built at the interfaces[38], thus leading to an efficiency increase in a fresh device for a short period. The TEA-based device shows a reduced EQE overshoot as 129% and an increased $T_{50}$ of ~50 min. In contrast, the PPT' device exhibits an enhanced operational lifetime as $T_{50}$ = ~30 h with a small overshoot of 102%. For PPT device, a remarkably elongated $T_{50}$ of ~220 h without any overshoot were observed. The EL spectra of PPT device did not show obvious shift during this long-term operation (Supplementary Fig. 46). Next, we evaluated the PPT device lifetime at increased current densities (Fig. 4h). The devices exhibit $T_{50}$ of 30.9, 7.1, 2.8 h at a constant current density of 1, 5, 12 mA/cm$^2$ with a corresponding initial luminance of 10, 35, 100 cd/m$^2$; respectively. These results are comparable to and even better than the existing quasi-2D perovskite LEDs (Supplementary Table 2). We further characterized degraded devices using ToF-SIMS. The PPT-based devices exhibit significantly suppressed iodine diffusion compared with the BA-based devices (Supplementary Figs. 47, 48). We acknowledge that more efforts are still required to boost the operational stability to the level of commercialization. For instance, other issues in terms of charge injection balance, electrochemical redox reactions in perovskites, as well as joule heating effect that may lead to the degradation of perovskite LEDs also need to be seriously considered and addressed in the future[39].

## Wavelength tunability and spectral stability

With better control over phase distribution, here we tune the EL emission across a range of spectra from 666 to 740 nm (Fig. 5a) utilizing precursor solutions with varied <$n$> numbers and mixed halide

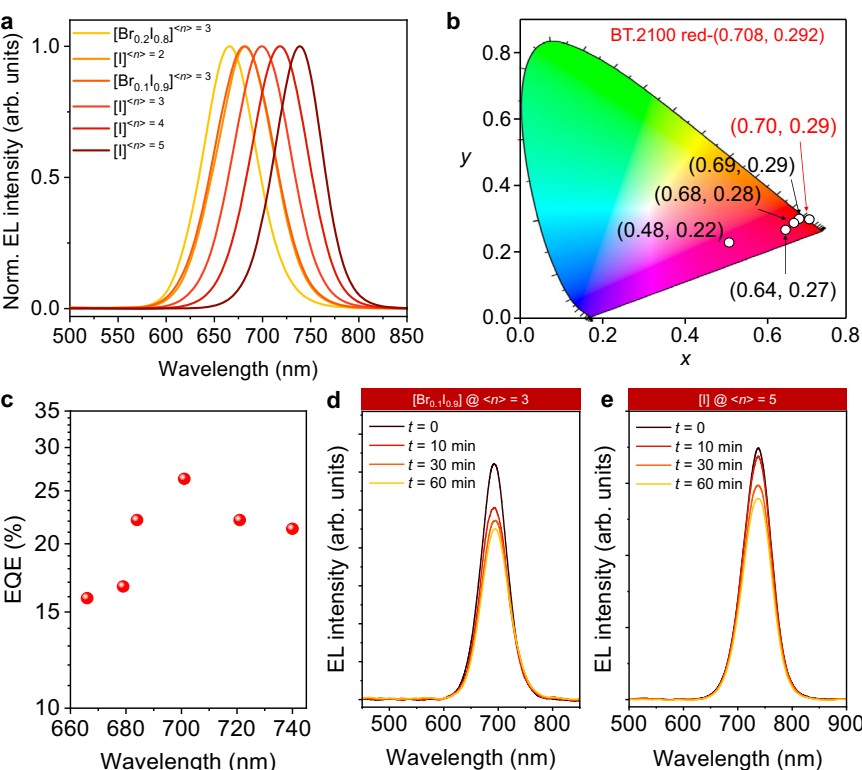

**Fig. 5 | Wavelength tunability and spectral stability. a, b** EL spectra (**a**) of PPT based quasi-2D perovskite LEDs with different compositions and the corresponding CIE coordinates (**b**). Typically, <$n$> = 3 represents that the stoichiometric ratio of precursor solution follows the general chemical formula of (PPT)$_2$FA$_{n-1}$Pb$_n$I$_{3n+1}$ with a nominal $n$ = 3. **c** The relationship of EQEs against EL peaks. **d, e** Time evolution tracking of EL spectra for LEDs fabricated from <$n$> = 3 with 10% Br + 90% I (**d**), and <$n$> = 5 with 100% I (**e**) qausi-2D perovskites. The current density was maintained constant at 1 mA cm$^{-2}$ during the measurement. The extraordinary spectral stability, where the EL spectra did not show obvious shift for both cases under long-term operation, demonstrates the great suppression of halide segregation and phase disproportionation, respectively.

contents (Supplementary Table 3). The Commission Internationale de l'Eclairage (CIE) coordinates of different emission wavelengths are plotted in Fig. 5b. Among them, high-purity red emission was obtained with a peak EQE of 22.1% (Fig. 5c) and a (CIE) coordinates of (0.70, 0.29), which matches well with the ITU-R Recommendation BT.2100 color space of pure red color. The device performance of quasi-2D perovskite LEDs with different emission wavelengths are shown in Supplementary Figs. 49, 50 with their peak EQEs summarized in Fig. 5c. The different EQE values at various emission wavelengths are probably due to distinct energy transfer pathways and varied defect densities among different $n$-phases, highlighting the critical role of phase distribution control. Importantly, the time evolution tracking of EL spectra (Fig. 5d, e) show extraordinary spectral stability in terms of suppressed $Br^-/I^-$ segregation and phase disproportionation. This is significantly improved compared with other LEDs based on quasi-2D or mixed halide perovskites[40,41].

## Discussion

In summary, we have demonstrated that suppressed phase disproportionation leads to highly efficient and wavelength-tunable quasi-2D perovskite LEDs with a peak EQE of 26.3% and good stability and reproducibility. To the best of our knowledge, this is the highest efficiency reported so far for a quasi-2D halide perovskite LED in the red region. We have found that ionic transport is an important step of solid-phase disproportionation reaction in quasi-2D perovskites, and the key functions of those bulky organic conjugated ligands are to stabilize the perovskite kinetic products and inhibit ion diffusion. Furthermore, we have shown that the devices exhibit wide wavelength tunability with extraordinary spectral stability. We believe this molecular engineering strategy could be expanded to other perovskite-based optoelectronic devices. This work also suggests that there is a bright future in transforming perovskite LED technology into real-world applications[42,43].

## Methods

### Chemicals and reagents

Organic solvents, including anhydrous $N,N$-dimethylformide (DMF), chlorobenzene (CB), ethanol, and solid chemicals, including lead iodide ($PbI_2$), Poly-TPD, PVP (Mw = 29 k, 40 k, 360 k, 1300 k), potassium acetate (KAc), formamidinesulfinic acid (FSA), muconic acid, 1,3,5-tris(1-phenyl-1H-benzimidazol-2-yl)benzene (TPBi), LiF were purchased from Sigma Aldrich. Formamidinium iodide (FAI), butylammonium iodide (BA), phenethylammonium iodide (PEA) were purchased from Greatcell Solar. All the above chemicals were used as received. More details on the synthesis of bulky organic conjugated ligands[25], PPT', PPT, TEA, 2 P, 2 T and 3 T can be found in the Supplementary Information.

### Fabrication of perovskite films

A precursor solution was firstly prepared by mixing 50 mM DMF stock solution of $(L)_2PbI_4$ (L = BA, TEA, PPT', PPT) with 50 mM DMF stock solution of $FAPbI_3$ (FAI: $PbI_2$ = 1.2:1) inside the nitrogen-filled glovebox with different volume ratios. For instance, PPT based $<n> = 3$ precursor solution was prepared by mixing 50 mM $(PPT)_2PbI_4$/DMF and $FAPbI_3$/DMF with a volume ratio of 1 to 2. After that, 1 mol% (relative to $Pb^{2+}$) of FSA or muconic acid was added to the above mixtures (Supplementary Fig. 51), which was stirred at 45 °C for 2 h. The solution was then filtered through a 0.22 μm PTFE syringe filter for further use. Note, FSA was added for defects passivation in all the thin film and device studies, and muconic acid was introduced for device stability studies. Bare Si/$SiO_2$ wafer, quartz, glass slides or indium-doped tin oxide (ITO) coated glass substrates were cleaned by ultrasonication in detergent, de-ionized (DI) water, acetone, and isopropanol for 15 min; then dried with dry air. The substrates were treated with UV-ozone for 20 min, and then transferred into a glove box for spin coating. The above precursor solution was spin-coated onto the pre-cleaned

substrates at 4000 rpm for 60 s, followed by thermal annealing on a hot plate at 100 °C for 10 min. These obtained films can be used for further characterizations.

### LED device fabrication

The pre-cleaned ITO substrates were treated by ultraviolet-ozone for 20 min to make the surface hydrophilic and then transferred into a nitrogen-filled glove box. 8 mg/mL Poly-TPD in chlorobenzene solution was subsequently spin-coated onto the ITO substrate at 2000 rpm for 60 s and baked at 150 °C for 20 min. Next, a solution of PVP in ethanol (5 mg/mL) was spin-coated onto the Poly-TPD surface at a speed of 2000 rpm for 45 s followed by annealing at 100 °C for 10 min. Afterwards, a saturated DMF solution of KAc was spin-coated on top of PVP at 3000 rpm for 30 s to decrease the thickness of interlayer and also passivate interfacial defects, which was then annealed at 100 °C for 10 min. The quasi-2D perovskite films were prepared by spin-coating the precursor solutions onto the above substrates at 4000 rpm for 60 s and annealing at 100 °C for 10 min. Subsequently, TPBi (60 nm), LiF (1.2 nm) and an Al (100 nm) anode were deposited by thermal evaporation under high vacuum ($<2 \times 10^{-6}$ mbar). The active area of the devices was defined to be 0.04 cm² by the overlapping area of the ITO and Al electrodes.

### Characterizations

**Nuclear magnetic resonance (NMR) spectra.** NMR spectra were acquired at room temperature using a Bruker AV 400-MHz spectrometer with $CDCl_3$ or DMSO-$d_6$ as the solvent and tetramethylsilane (TMS) as an internal standard. Chemical shifts of $^1H$ NMR and $^{13}C$ NMR signals were reported as values (ppm) relative to the TMS standard.

**Mass spectra.** High resolution mass spectrometry was acquired in positive Electrospray mode (ESI) on an LTQ Orbitrap XL instrument (Thermo Fisher Scientific).

**Single-crystal XRD analysis.** Single crystals of $(PPT')_2PbI_4$ and $(PPT)_2PbI_4$ were analyzed using a Bruker Quest diffractometer with kappa geometry, an I-μ-S microsource X-ray tube, a laterally graded multilayer Göbel mirror for single crystal monochromatization, an area detector (Photon2 CMOS). Data collections were conducted at 150 K with Cu Kα radiation ($\lambda = 1.54178$ Å).

**UV-vis absorption spectra.** Thin film absorption spectra were recorded on an Agilent UV-Vis-NIR Cary-5000 spectrometer in transmission mode.

**Photoluminescence (PL) spectra.** Steady state PL spectra were obtained with an Olympus microscope system (BX53) integrated with an X-CITE 120Q UV lamp. The filter cube contains a bandpass filter (330–385 nm) for excitation, a dichroic mirror (cutoff wavelength, 400 nm) for light splitting, and a 420 nm long-pass filter for emission collection. The collected PL signals were analyzed by a spectrometer (SpectraPro HRS-300). In-situ PL measurements were carried out in a nitrogen-filled heating stage (INSTEC, mK2000).

**Photoluminescence quantum yield (PLQY).** The PLQYs were obtained by a three-step technique with a home-designed system, which consists of a continuous-wave laser (375 nm), an integrating sphere, optical fiber and a spectrometer[44].

**Kelvin probe force microscopy (KPFM).** The KPFM topography and CPD images were obtained with Asylum Research Cypher ES Environmental AFM in the air.

**Powder XRD.** Quasi-2D perovskite thin films were spin-coated on pre-cleaned glass substrates for XRD measurements. Thin film XRD was

measured with a Rigaku Smart Lab (Cu Kα, $\lambda = 1.54056\,\text{Å}$) in Bragg Brentano (BB) mode.

**Grazing incidence small angle X-ray scattering (GISAXS).** GISAXS and in-situ GIWAXS spectra were collected at beamline 7.3.3. at the Advanced Light Source at Lawrence Berkeley National Lab utilizing an incident angle of 0.2° and wavelength of 1.24 angstrom (energy 10 keV). 2D spectra were taken with a Pilatus 2M-2D detector and integrated to reduce to 1D with the Igor Pro NIKA GIWAXS software[45]. Raw data was further processed and visualized using Xi-Cam[46].

**Time-resolved PL (TRPL) measurements.** A home-built PL microscope was used to conduct TRPL measurements. A picosecond pulsed excitation beam with a wavelength of 447 nm was focused onto the sample by using a 40× objective (Nikon, NA = 0.6). The PL emission was collected by the same objective, and then guided to a single photon avalanche diode (PicoQuant, PDM series) with a single photon counting module (PicoQuant).

**Transient absorption (TA) spectroscopy.** Transient absorption spectra of quasi-2D thin films were measured by a home-built femtosecond pump-probe system described in a previous publication[30]. Briefly, part of a 1030 nm fundamental Yb:KGW laser (PHAROS, Light Conversion Ltd.) output was focused onto a YAG crystal to generate the broad-band probe, and the rest of the fundamental laser was sent to an optical parametric amplifier (OPA, ORPHEUS-Twins, Light Conversion Ltd.) to generate wavelength tunable pump pulses. An optical chopper (MC200B, Thorlabs) was used to modulate the pump beam with a frequency of 195 Hz. A linear stepper motor stage (Newport) was utilized to delay the probe relative to the pump beam. Both pump and probe beams were focused on the sample with a 5× objective (N.A. = 0.1, Olympus). The pump induced change in the probe transmission was collected by a 20× objective (N.A. = 0.45, Olympus) and detected by an array detector (Exemplar LS, B&W Tek).

**Scanning electron microscopy (SEM).** SEM images were taken using a FEI Teneo VS SEM at 10 kV and 0.10 nA using a back-scattered electron detector.

**Ultraviolet photoelectron spectroscopy (UPS) measurement.** The UPS measurements were conducted in a PHI 5600 analysis system (base pressure $3 \times 10^{-10}$ mbar) with a hemispherical electron energy analyzer equipped with a multichannel plate detector. The spectra were collected with a H Lyman-α photon source (E-LUX 121, photon energy of 10.2 eV) and a 5.85 eV pass energy while the samples were biased at negative 5 V.

**Optical constant measurement.** The optical properties of Poly-TPD, PPT based quasi-2D perovskite, TPBi, and Al were obtained using a J. A. Woollam V-Vase UV-vis-NIR spectroscopic ellipsometer from 500 to 800 nm. The incident angle for all the measurements were at 50° and 70°. To conduct the optical characterization, the materials were coated on a microscopic glass substrate. The imaginary part of the complex dielectric function for Poly-TPD, perovskite and TPBi were fitted using a Tauc-Lorentz model and the bandgap was matched to the available literature. The measured data was corrected using the built-in backside model correction to account for the transparent substrate.

**Device characterizations.** All LED devices were characterized at room temperature in a nitrogen-filled glovebox without encapsulation. The current density versus voltage characteristics were measured using a Keithley 2450 source-measure unit. The devices were swept from zero bias to forward bias at a rate of 0.2 V s⁻¹. In our lab, a 100 mm PTFE integrating sphere coupled with a spectrometer (Enli Technology, LQ-100X) were used for the measurements of radiance,

electroluminescent spectra, EQE, CIE, and operational stability. The whole system is calibrated by a NIST-traceable standard QTH lamp. From this step, one can get the full-spectrum spectral response (μW) of the systems. The operational stabilities of LEDs were measured at constant current densities. In addition, the LED devices were cross-checked at Taiwan (Prof. Guo's group at NCKU), where the EL spectra were examined by PR655 (SpectraScan, Photo Research, Inc.), the emitted photons from the devices were recorded by photodiode (Si photodiode, $10 \times 10$ mm S2387-1010R, Hamamatsu Photonics, UK) and read by Keithley Picoammeter 6485. The EQE values were calculated based on photocurrent and EL spectrum component by following the methods described in a previous publication[47].

**Electrochemical impedance spectroscopy (EIS) measurements.** The frequency dependent capacitance characterization was carried out on complete devices using a Versa STAT electrochemical workstation (Ametek). Devices were measured using a 30-mV perturbation sinusoidal AC voltage with frequency ($f$) swept from $10^6$ to $10^2$ Hz under zero DC bias voltage.

**Galvanostatic tests.** 5 nm Cr and 20 nm Au layers were sequentially deposited on perovskite by thermal evaporation. The channel was defined by an Al wire with diameter of 20 μm serving as shadow mask. The samples were loaded on Linkam LTS350 stage purged with dry nitrogen. DC bias was applied, and current was measured at dark using Keithley 4200-SCS parameter analyzer with 4200-PA remote pre-amplifier. The voltage was turned on after device initialization and turned off as the ion accumulation was saturated, while the current over time was monitored. The samples were relaxed over 30 min after each test till they recovered from the polarized state.

**Time-of-flight secondary ion mass spectrometry (ToF-SIMS).** Positive high mass resolution depth profile was performed using a TOF-SIMS NCS instrument, which combines a TOF.SIMS5 instrument (ION-TOF GmbH, Münster, Germany) and an in-situ Scanning Probe Microscope (NanoScan, Switzerland) at Shared Equipment Authority from Rice University. The analysis field of view was $100 \times 100$ μm² ($Bi_3^+$ @ 30 keV, 0.3 pA) with a raster of 128 by 128 along the depth profile. A charge compensation with an electron flood gun has been applied during the analysis. An adjustment of the charge effects has been operated using a surface potential. The cycle times was fixed to 100 μs (corresponding to m/z = 0–911 a.m.u mass range). The sputtering raster was $450 \times 450$ μm² ($Cs^+$ @ 1 keV, 45 nA). The beams were operated in non-interlaced mode, alternating 1 analysis cycle and 1 frame of sputtering (corresponding to 1.55 s) followed by a pause of 2 s for the charge compensation. The $MCs_n^+$ (n = 1, 2) depth profiling has been also used for improving the understanding of the data. This is a useful method, mainly applied to quantify the alloys but also to identify any ion compounds. The cesium primary beam is used for sputtering during the depth profile and permits to detect $MCs^+$ or $MCs_2^+$ cluster ions where M is the element of interest combined with one or two Cs atoms. The advantages of following $MCs^+$ and $MCs_2^+$ ions during ToF-SIMS analysis include the reduction of matrix effects and the possibility of detecting the compounds from both electronegative and electropositive elements and compounds. All depth profiles have been point-to-point normalized by the total ion intensity and the data have been plotted using a 20-points adjacent averaging. Both normalization and smoothing have permitted a better comparison of the data from the different samples. The depth calibrations have been established using the interface tool in SurfaceLab version 7.2 software from ION-TOF GmbH to identify the different interfaces and based on the measured thicknesses using the surface profiler to obtain a line scan of the craters with the in-situ SPM by contact scanning.

## Simulations

**Molecular dynamics (MD) simulations.** The modified MYP model was used for all simulations as described previously[25]. LAMMPS and PLUMED were used to perform the MD simulations[48,49]. All simulations used 1 fs integration timestep and periodic boundary conditions (PBC). Long-range electrostatics were modeled using the particle-particle-particle-mesh (PPPM) algorithm and Lennard-Jones interactions were truncated at 15 Å. The initial structures of the perovskites were generated by constructing representative unit cells of ideal perovskites with the bulky organic ligands placed at the surface. The simulation was first relaxed in the NVE ensemble with restrained atomic displacements of 0.01 Å per timestep for 50 ps, followed by a 1 ns NPT simulation with the Nose-Hoover thermostat and barostat. The boundary in the y-direction, which by construction is parallel to the bulky organic ligands, is set at the end of the surface bulky organic ligands so that the tails of organic ligands from different sides will interact with each other due to the PBC and thus form stacked structures. During the NPT equilibration, the barostat was only applied to x and z-direction, which are normal to the bulky organic ligands. After the equilibration, steered molecular dynamics (SMD) were used to calculate the free energy profiles[50]. In SMD, we use a spring constant of 100 kcal·(mol·Å)$^{-1}$ and a constant velocity of 0.036 Å/ps to steer the target diffusing halide through the organic bulk ligands. For each ligand, five independent simulations with randomly chosen halides from the lattice are conducted to obtain the average value for the free energy profile.

**Optical simulation.** The optical simulation was performed on a three-dimensional model using the RF module of COMSOL Multiphysics 5.5. A classic electric point dipole is placed in the middle of perovskite layer, which is assumed to be isotropic. A perfect matching layer (PML) and full-wave incidence are applied to the modelling configuration. The power fraction radiated to each wavevector region (outcoupled, substrate, waveguiding, and plasmonic) was simulated by using the experimentally obtained optical constants (refractive index and absorption coefficient) of each layer retrieved from ellipsometry measurements. The thicknesses of the ITO, Poly-TPD, perovskite, TPBi, and Al are set to be 120, 30, 20, 60, and 100 nm, respectively.

## Data availability

Crystallographic data for the structure reported in this Article have been deposited at the Cambridge Crystallographic Data Centre under deposition number CCDC 2166731 [(PPT')$_2$PbI$_4$, https://www.ccdc.cam.ac.uk/structures/Search?access=referee&ccdc=2166731&Author=Zeller] and 2113526 [(PPT)$_2$PbI$_4$, https://www.ccdc.cam.ac.uk/structures/Search?access=referee&ccdc=2113526&Author=Zeller]. Copies of the data can be obtained free of charge via https://www.ccdc.cam.ac.uk/structures/. All other data supporting the findings of this study are available within the Article and its Supplementary Information.

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

## Acknowledgements

This work is supported by the National Science Foundation (Grant No. 2131608-ECCS). Z.Z. and H.G. acknowledge the support from the National Science Foundation (2110814-EPM). L.J. and L.H. acknowledge the support for spectroscopy measurements from the U.S. Department of Energy, Office of Science, Office of Basic Energy Sciences under Award No. DE-SC0022082. H.R.A. and K.R.G. acknowledge the support for UPS measurements from the U.S. Department of Energy, Office of Science, Office of Basic Energy Sciences under Award No. DE-SC0018208. X.M. acknowledges the financial support by the National Natural Science Foundation of China (No. 11774188). A.B. and S.C. acknowledge the support from the ONR MURI (N00014-21-1-2026). ToF-SIMS analysis was carried out with support provided by the National Science Foundation CBET-1626418 and this work was conducted in part using resources of the Shared Equipment Authority at Rice University. We thank Dr. Shuchen Zhang and Qian Lu for the discussions and help with device fabrications, respectively.

## Author contributions

L.D. conceived the idea and supervised the project. K.W. carried out the materials synthesis, structural characterizations, device fabrications, and data analysis. Z.L. and B.M.S. performed molecular dynamics simulations and data analysis. Z.Z. and H.G. conducted galvanostatic tests of charging current decay. L.J. and L.H. carried out the ultrafast spectroscopy measurements. K.M. carried out electrochemical impedance spectroscopy and KPFM measurements. A.H.C. and C.Z. performed GISAXS and in-situ GIWAXS measurements. H.R.A. and K.R.G. carried out UPS measurements and data analysis. Y.G. and Z.W. performed organic ligands synthesis and structure characterization. J.P. grew the single crystal and solved the structures. B.P.F. performed SEM characterizations. X.M. carried out optical simulations. S.C. and A.B. conducted optical constant measurements. Z.C., T.T., and Y.Y. conducted ToF-SIMS measurements and data analysis. T.D. and T.G. carried out efficiency cross-check, provided insightful discussions, and participated in data analysis and manuscript preparation. K.W. and L.D. wrote the manuscript. All authors discussed the results and revised the manuscript.

## Competing interests

The authors declare no competing interests.
