## [Peer Review File · Nature Communications]

Suppressing Phase Disproportionation in Quasi-2D Perovskite Light-Emitting DiodesEditorial Note: This manuscript has been previously reviewed at another journal that is not operating a transparent peer review scheme. This document only contains reviewer comments and rebuttal letters for versions considered at *Nature Communications*.

REVIEWERS' COMMENTS

Reviewer #1 (Remarks to the Author):

The authors presented novel organic spacers for highly efficient and stable quasi-2D perovskite emitters enabled by suppressed phase disproportionation and ionic transport. With precise in-situ analysis well correlated with molecular dynamics simulation, the effect of systemically designed new ligands was successfully addressed.

I agree with the revisions made by the authors. The quality of the manuscript has been improved. I recommend the acceptance of the manuscript.

Reviewer #2 (Remarks to the Author):

I have read through all the comments and the point-by-point responses. The authors have made a substantial effort to address the reviews' comments. And the revision meets the standard for publication now. I recommend that this manuscript can be accepted without any other change.

Reviewer #3 (Remarks to the Author):

The reviewer thanks the authors for making an effort to revise the manuscript again and solve my worries. Comparing durability results of different LED architectures is very different and should be handled with care. By the way the manuscript quality has reached a level for publication in nature communications.

Responses to Reviewers' Comments (*changes have been highlighted in yellow*)

Reviewer #1 (Comments for the Author):

The authors presented novel organic spacers for highly efficient and stable quasi-2D perovskite emitters enabled by suppressed phase disproportionation and ionic transport. With precise in-situ analysis well correlated with molecular dynamics simulation, the effect of systemically designed new ligands was successfully addressed.

I agree with the revisions made by the authors. The quality of the manuscript has been improved. I recommend the acceptance of the manuscript.

Response: Thank the reviewer for reviewing our manuscript and appreciating the importance of our work in the field. We are very grateful for those constructive comments and suggestions given by the reviewer, which have guided us to further improve the quality of our work.

Reviewer #2 (Comments for the Author):

I have read through all the comments and the point-by-point responses. The authors have made a substantial effort to address the reviews' comments. And the revision meets the standard for publication now. I recommend that this manuscript be accepted without any other change.

Response: We would like to thank the reviewer for reviewing our manuscript again and providing the positive comments. We are very grateful for those constructive comments and suggestions given by the reviewer, which have guided us to further improve the quality of our work.

Reviewer #3 (Comments for the Author):

The reviewer thanks the authors for making an effort to revise the manuscript again and solve my worries. Comparing durability results of different LED architectures is very different and should be handled with care. By the way the manuscript quality has reached a level for publication in nature communications.

Response: We thank the reviewer for reviewing our manuscript again and providing the positive feedback. We are very grateful for those constructive comments and suggestions given by the reviewer, which have guided us to further improve the quality of our work.